# Piezoresistive Sensor Based on Micrographite-Glass Thick Films [note 1]

**DOI:** 10.3390/s22093256

**Published:** 2022-04-24

**Authors:** Osvaldo Correa, Pompeu Pereira de Abreu Filho, Stanislav Moshkalev, Jacobus Swart

**Affiliations:** 1Faculty of Electrical Engineering and Computing—FEEC, University of Campinas, Campinas 13083-852, SP, Brazil; osvaldocorrea50@gmail.com (O.C.); jacobus@unicamp.br (J.S.); 2Center for Semiconductor Components and Nanotechnologies—CCSNano, University of Campinas, Campinas 13083-870, SP, Brazil; pompeu@sigmabbs.com.br

**Keywords:** piezoresistive material, thick film, conductive paste, micrographite particles, glassy matrix

## Abstract

A new Pb-free glass containing several oxides (Bi_2_O_3_, B_2_O_3_, SiO_2_, Al_2_O_3_ and ZnO) with sintering temperature reduced down to 600 °C has been developed for applications in a piezoresistive pressure sensor. Using this low sintering temperature glass, it was possible to fabricate micrographite-based pastes and piezoresistive films without losses of graphitic material during the sintering. Good adherence of the films onto alumina substrates was observed and attributed in part to the reactions of ZnO and Bi_2_O_3_ with alumina substrates. Piezoresistive films with uniformly distributed micrographite particles were produced using sodium carboxymethyl cellulose (NaCMC) in aqueous solutions during the preparation of pastes. NaCMC plays a decisive role in interactions between micrographite particles and glassy matrix, providing good wettability of glass powder particles and homogeneous distribution of MG particles in the pastes. Finally, excellent repeatability of the sensor response to the applied deformations was verified in cycling experiments when the sample was submitted to 1000 load/release cycles. These results demonstrated very high stability of the sensor response (within ±1%), and also evidenced high stability of the film under the cyclic strain loads and good film adherence to the substrate.

## 1. Introduction

The piezoresistive thick film technology was introduced decades ago for the production of hybrid circuits, resistors and components for various industrial segments, such as automotive, military, aerospace, information technology, computing and instrumentation [1,2,3,4,5]. Nowadays, thick film technology has been spreading in numerous industrial sectors and involves new classes of materials [6,7,8,9,10,11]. Usually, piezoresistive devices employ various types of ceramics or glass due to their excellent electromechanical coupling properties and robustness; however, in recent years some other materials such as polymer-based piezoelectric nanocomposites [6,9,11] have emerged as potential alternatives in particular for applications in wearable devices due to high flexibility of these materials. Specific requirements for the application, such as enhanced mechanical flexibility, durability, tunability of properties, cost, compatibility with the substrate, and ease of processability among others, determine the type and composition of the piezoresistive or piezoelectric material to be employed in the device. 

In the present work, new Pb-free glass-based piezoresistive material was developed, looking for application in low-cost industrial pressure sensors with enhanced durability and stability. Thick film technology for pressure sensor devices based on ceramics and glass is largely used in automotive vehicles, as air conditioning pressure sensors, fuel pumps pressure sensors, common rail pressure sensors, brake air reservoirs in brake systems, fuel level sensors, wearable devices, etc. Not less important, the automation industry also applies thick film technology in ceramic alumina substrates for insulated-gate-bipolar transistor modules, high current diode bridge rectifier modules, and ceramic micro-electromechanical devices to manufacture high-pressure sensors for pneumatic and hydraulic systems.

Thick film technology for piezoresistive devices basically is based on three main components: (i) functional phase, which is also called a conductive phase, usually composed of metals or metal oxides such as RuO_2_, Bi_2_Ru_2_O_7_ or Pb_2_Ru_2_O_6_ [4]; (ii) a glass matrix as a permanent binder, such as Pb or Bi borosilicate, which allows the adherence of glass matrix to the substrate; and (iii) a vehicle that is responsible for the right rheological behavior of the paste, concerned with the fabrication process where usually ethyl-cellulose and α-terpineol are used as thickener and solvent, respectively. In addition, small amounts of dispersants and plasticizers are added to obtain the correct paste rheology and easier printability by screen-printing on substrates, providing better resolution in fabricated structures. 

To replace extremely expensive ruthenium oxides, the piezoresistive film process presented here uses micrographite (MG) particles (mean lateral size of 3 microns) as a conductive phase. The permanent binder (glassy matrix) is composed of a special formulation Pb-free glass with a low-temperature soft point (T_s_), and the organic phase (vehicle) is an aqueous solution of sodium carboxymethyl cellulose (NaCMC). To avoid losses of MG by oxidation in air atmosphere furnace, the thermal processing (sintering) consists of heating up to the dwell temperature to near 600 °C, keeping at this temperature for 10 min and then, cooling down to the environmental temperature. 

It is important to emphasize that to the best of our knowledge, this is the first report on the micrographite/glass-based films for low-cost piezoresistive sensor applications, with preliminary results recently published elsewhere [12], see also Appendix A. 

The replacement of ruthenium oxide by MG is justified considering the price and scarcity of metallic Ru (Ru0) and its excellent electrical, thermal and mechanical properties of micrographites. Ru0 is a platinum group metal (8B), with low abundance in the earth’s crust [13]. The high demand for ruthenium has kept its price between USD 6430/kg and USD 11,254/kg in recent years [14,15]. The use of ruthenium oxides had been justified by their thermal stability, resistance to oxidation and low electromigration. Note that even considering that the commercialized pastes have an average of only 15 to 20% of ruthenium oxides, their prices are highly impacted by the costs of Ru0. For example, the DP 2041 (DuPont) paste has been quoted at USD 8700/kg. The shortage of Ru0 also does not encourage industrial R&D investments in products based on Ru0. Therefore, besides the much lower cost and higher availability, micrographites have other significant advantages, such as high thermal resistance, and the absence of electromigration at high temperatures.

## 2. Materials and Methods

The micrographite powder (Micrograf 99503UJ) was supplied by Nacional de Grafite (Brazil) with an average accumulated diameter at 50% (d50) close to 3 μm, and the mean thickness is in the order of magnitude smaller (100–200 nm). The sodium carboxy methyl cellulose (NaCMC) was supplied by Denver (Brazil), with pH=6.9 (4% solution) and a viscosity coefficient of 90 cP (4% solution, 25 °C). The other analytical grade reagents were: Bi_2_O_3_, ZnO, SiO_2_, Al_2_O_3_, H_3_BO_3_, Ag/Pd conductive Paste DuPont 7484R and alumina plates with dimensions of 50 × 25 mm and thickness of 630 µm.

Several different Pb-free glass compositions were tested here, in order to reach a balance between various mechanical and electrical characteristics, such as stiffness, adherence to the substrate and piezoresistivity. Four main formulations are shown in Table 1. In these compositions, bismuth and boron oxides are responsible for the reduction in the sintering temperature [16,17] while silicon, zinc and aluminum oxides were added to improve the film stiffness and adherence to the alumina substrate [18,19].

The preparation of glass powders composed of various combinations of Bi_2_O_3_, B_2_O_3_, SiO_2_, Al_2_O_3_ and ZnO was performed under ambient conditions by the mixing of glass components. The solid mixture of reagents was homogenized and transferred to an alumina crucible. Then, the crucible was transferred to a furnace at room temperature and heated up to 900 °C with a rate of 10 °C/minute and 25 min at a final (dwell) temperature. After melting, the liquid mixture was poured quickly into cool deionized water, (a process known as the quenching process), see Figure 1A, followed by a first grinding step in a mortar grinder Retsch. After the grinding, the glassy matrix was sieved through a 200-mesh sieve (74 µm size), obtaining yellowish powder as shown in Figure 1B. After this, a second grinding step was performed using a high-energy ball mill (alumina balls) to produce a micro particle powder with a mean diameter of 5.94 µm (Figure 1C). The tapped density of the milled glassy matrix after the second grinding was 3.38 g/cm^3^.

The conductive Ag/Pd film was screen-printed on the alumina substrate to form conductive tracks for electrical connections of MG conductive film. This film was made using the AgPd paste (DuPont 7484R), applied on alumina substrate in the presence of tape masks, to improve the device shape definition and thickness. After printing the paste, the deposited films were kept at rest for 15 min., and afterward, the tape masks were removed. Then, the films were heated initially to 150 °C (heating speed 10 °C/min.) for drying and then kept for 10 min at 850 °C, for sintering. This thermal processing was carried out using an oven EDG, model FCVI-I. For electrical contacts, copper wires (AWG 23) were soldered with Sn/Pb solder (63/37) over Ag/Pd tracks.

The piezoresistive micrographite-based pastes were prepared using solid mixtures with a mass ratio of MG/glass powder of 1:10, adding the aqueous solution of sodium carboxymethyl cellulose (NaCMC) of 2% (*w/w*). The mass ratio of MG: NaCMC in the paste was between 1:0.06 and 1:0.09. The mixtures, after adding the NaCMC solution, were homogenized and then immediately distributed evenly over alumina substrates using a squeegee.

Piezoresistive device responses (change in resistance, ∆R) to a mechanical strain were evaluated for prepared films using a cantilever clamped beam geometry (Figure 2). In this approach, the beam strain is generated by a calibrated force (load) applied at the free (not clamped) end of the cantilever in the normal direction. The maximum vertical deflection of the cantilever edge (at the point of F force application) in tests varied from a few to a few tens of microns. In this case, the change in the film electric resistance under strain and corresponding film deformation together with the alumina substrate is mostly due to changes in the contacts between conductive micrographite particles in the film. These changes result in an increase in the total resistance, typical for piezoresistive devices. The ends of the conductive films were deposited over Ag/Pd film tracks that served as a conductive path, and the copper wires (AWG 23) were soldered on the film track to enable electrical contacts in electrical measurements using a multi-meter Agilent 34401A. 

Note that the mechanisms of electrical conduction in the piezoresistive films based on MG particles are still not well understood and need further studies. Discussion in the literature for other systems with conductive particles of different origins, including RuO2, shows that the standard percolation theory of transport universality was proved only in a limited number of experiments on real disordered composites [20]. In these experiments, the tunneling-percolation model was applied to carbon-polyvinyl chloride composites where the inter-grain tunneling was found to be the principal mechanism of transport [21]. An attempt to develop a theoretical model to relate the resulting gauge factors in piezoresistive sensors to different types of nanofillers has been reported in a recent work [22]. It is thus reasonable to suggest that for the MG-based piezoresistive films a tunneling-percolation model is also valid.

The MG-based films were made by screen printing the paste on alumina substrates. Before deposition, tape masks were prepared on the alumina substrate. After resting for about 15 min at ambient conditions, the masks were removed, and the films were heated up to 600 °C (rate of 10 °C/min and 10 min at final temperature) in air atmosphere.

## 3. Results

In the initial stage of experiments, several glass frit compositions were tested, in order to optimize their mechanical characteristics such as stiffness and adherence to the substrate. The presence of NaCMC was crucial, allowing the formation of homogeneous MG-glass pastes easily distributed over the alumina substrate. In the absence of NaCMC (pastes containing only MG filler, glass powder and water), the formation of continuous uniform films with good adherence to the substrate was not possible, indicating that NaCMC has a decisive role in the interactions between MG particles and glass powder. The NaCMC chains dissolved in water are amphiphilic, and they form van der Waals bonds with the initially hydrophobic surfaces of MG particles [23]. This results in deep modification of graphitic surfaces, providing their partial hydrophilicity due to hydroxyl groups of the NaCMC chains bonded to the graphitic surface. Partial intercalation of MG particles by NaCMC molecules followed by their exfoliation also may occur in these interactions [24]. These hydrophilic micrographite-NaCMC particles are responsible for the wettability of the resulting pastes and good adherence to the alumina substrates. Further, better adhesion of the films to alumina substrates was observed for compositions with MG: Glass mass ratio of 1:10, with the volume ratio Glass to MG of 0.6:1. The adhesion was tested by a manual force applied with a stainless still spatula at the film–substrate interface, and the film detachment was not possible due to strong film adhesion. Further evidence of strong film adhesion and hardness was obtained in the film abrasion tests with sandpapers, which demonstrated high abrasion resistance.

Furthermore, four main film formulations (see Table 1) were tested to optimize the film adhesion to the substrate under strain deformations. The strain was applied to films in a cantilever clamped beam geometry (Figure 2), with the maximum load of 1 kgF for 10 min, and then, after the load removal, the evolution of the film resistance was monitored for 2 h (or even for longer periods), see Figure 3.

In most formulations shown in Table 1, with an exception for the sample 102 (30Bi_2_O_3_/ 48B_2_O_3_/10SiO_2_/2Al_2_O_3_/10ZnO), long term changes in the film resistance after the strain removal were detected, usually consisting of several phases with the initial decrease in resistance followed by its increase with partial or full recovering of the initial resistance, in time periods up to tens of hours (not shown). The whole sequence of the resistance evolution includes at least three phases: (i) initial fast rise (during strain application), (ii) slow decrease after strain removal, and (iii) slow increase to recover the initial film resistance. This long-term evolution probably can be attributed to residual deformations and also to partial detachment of the films from the substrate during the application of strain, resulting in slow changes in the measured film resistance after the strain removal.

The largest deviations were detected for samples 65 (30Bi_2_O_3_/51B_2_O_3_/15ZnO/4SiO_2_) and 66 (30Bi_2_O_3_/45B_2_O_3_/15ZnO/10SiO_2_) with the resistance decreasing by 3.5%, and the largest deviation observed for the sample with higher SiO_2_ content. Sample 97 (30Bi_2_O_3_/55B_2_O_3_/15ZnO), not including SiO_2_, had better resistance stability (decreasing by 1.5%), but had lower film stiffness. The best results in terms of resistance stability were obtained for sample 102, where aluminum oxide was added. Alumina was reported to act as a component to avoid crystallization (devitrification) of glasses [25,26]. This effect also could be attributed to the difference in thermal coefficients of expansion (TCE) of the substrate and the film printed on it; that is, it was reduced with the addition of alumina to the glass. The difference in TCE of the film and substrate can induce residual strain in the film printed on the substrate after the final thermal treatment of the sensor, resulting in a slow resistance variation, especially after the application of a large external strain.

The typical results of electrical measurements under strain experiments with the load force applied up to 800 gF for the sample 102 with optimized composition are shown in Figure 4, where the piezoresistive effect for the prepared conductive film can be observed with an increase in the electrical resistance under the action of strain. A linear increase in the film resistance with the load was verified.

Further, to examine the repeatability of measurements, the sample was submitted to one thousand load/release cycles, with a load of 600 gF. The measurements of resistance were taken for each 10 load-release cycles period, so that a total of 200 readings (before and after the load, with a total number of 100 loads) were performed. The results are shown in Figure 5. Very high stability of the sensor response (4.75 ± 0.05 mV, i.e., within ±1%) can be seen. This result also evidences the very high stability of the film under the cyclic strain loads and good film adherence to the substrate. As applications for the automotive industry typically require a 2% error for a pressure sensor, the sensors based on the material developed here (with some further improvements) clearly have good potential for such applications.

Structural characterizations of the films were performed using scanning electron microscopy (SEM), with Dual-beam FIB/SEM Nova 200 Nanolab. Figure 6 shows the SEM image of the film surface where MG particles (grey color) are uniformly distributed in the glassy matrix. Some pores (darker areas) in the film surface can be also seen. The brighter areas correspond to non-conductive glassy matrix particles that are charged by the electron beam. The uniform distribution of MG particles in the glassy matrix and the presence of some pores can be also observed. Figure 7 shows a cross-section of the fractured film, with lower (left) and higher (right) magnifications. Tight contact between the film and the substrate without any voids can be seen, indicating very good adherence of the film. 

The surface morphology of the films was characterized by a 3D optical profilometer PS50 from Nanovea. Figure 8 shows a typical 3D image, with vertical and lateral scales of 6000 nm and 50,000 nm (6 and 50 µm), respectively, where multiple hillocks with heights and widths of a few microns can be seen. 

Figure 9 shows Raman spectra obtained for the micrographite samples before processing (A) and for the fabricated piezoresistive films including micrographite particles (B).

Minimal changes to the graphene characteristic lines (line D at ~1360 cm^−1^, line G at ~1580 cm^−1^ and line 2D, at ~2750 cm^−1^) can be observed. The ratio of intensities for D and G bands (I_D_/I_G_) is frequently used to characterize the density of defects in graphene samples [27], and small ratios show the high quality of graphene. Very small changes in the I_D_/I_G_ ratio after all the processing steps to fabricate the film, from 0.12 to 0.15, indicating that the high quality of the graphitic material was basically maintained. 

## 4. Discussion

Pb-free glass with sintering temperature reduced down to 600 °C containing several oxides (Bi_2_O_3_, B_2_O_3_, SiO_2_, Al_2_O_3_ and ZnO) was developed. Using this glass, it was possible to fabricate uniform micrographite-based piezoresistive films without losses of graphitic material. Good adherence of the films onto alumina substrates is attributed in part to the reactions of ZnO and Bi_2_O_3_ with alumina substrates likely due to the formation of ZnAl_2_O_4_ at the alumina–film interface. Raman spectra for the micro-graphite taken before and after sintering indicated minimal changes to the graphene characteristic lines, and, thus, confirming that the high quality of the graphitic material was maintained during the film preparation.

The 1:10 MG: glass matrix mass relation was found to be optimal to provide the enhanced adherence of the piezoresistive films on alumina substrates that serve as a base of the piezoresistive sensor. The piezoresistive characteristics of the films were tested in the mechanical strain experiments using a cantilever clamped beam geometry. These results showed high stiffness of the films, with full restoration of the film resistance after load removal, achieved for the optimized film composition. 

The use of sodium carboxymethyl cellulose (NaCMC) with its amphiphilic character was important to establish van der Waals bonds with micrographite particles transforming their initially hydrophobic surfaces into hydrophilic. This resulted in the formation of a composite material with micrographite readily dispersible in an aqueous medium, promoting the high wettability of glass powder particles and a final homogeneous paste. The role of NaCMC was decisive in obtaining high-quality piezoresistive films obtained with high-quality (low-defect density) micrographite. It was verified that pastes obtained in the absence of NaCMC did not show good adherence and were not stable when deposited over alumina substrates. 

Finally, excellent repeatability of the sensor response to the deformations was verified in cycling experiments when the sample was submitted to 1000 load/release cycles. These results demonstrated very high stability of the sensor response (within ±1%) and also evidenced high stability of the film under the cyclic strain loads and good film adherence to the substrate. As applications for the automotive industry typically require a 2% error for pressure sensors, the sensors based on the material developed here, with some further improvements in order to scale up the sensor fabrication, clearly have good potential for such applications. 

## 5. Conclusions

In summary, a new Pb-free glass containing several oxides with strongly reduced sintering temperature was developed. Using this glass, it was possible to fabricate uniform piezoresistive films containing micrographite flakes as a conductive component using a glass sintering process at 600 °C, for applications in a pressure sensor. Good adherence of the films onto alumina substrates, essential for the sensor application, was also confirmed in a series of repetitive load/release tests. Good adherence was attributed to the reactions of ZnO and Bi_2_O_3_ with alumina substrates likely due to the formation of ZnAl_2_O_4_ at the alumina–film interface. Very high stability of the sensor response to deformations (within ±1%) was verified in cycling experiments when the sample was submitted to 1000 load/release cycles. 

## Figures and Tables

**Figure 1 sensors-22-03256-f001:**
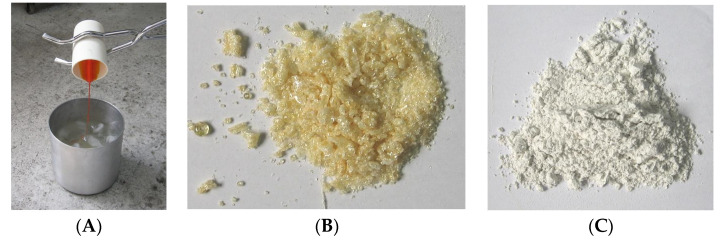
Main steps of frit powder preparation: (**A**) glass mixture cooling in water, (**B**) glass powder after first grinding, (**C**) glass powder after second grinding.

**Figure 2 sensors-22-03256-f002:**
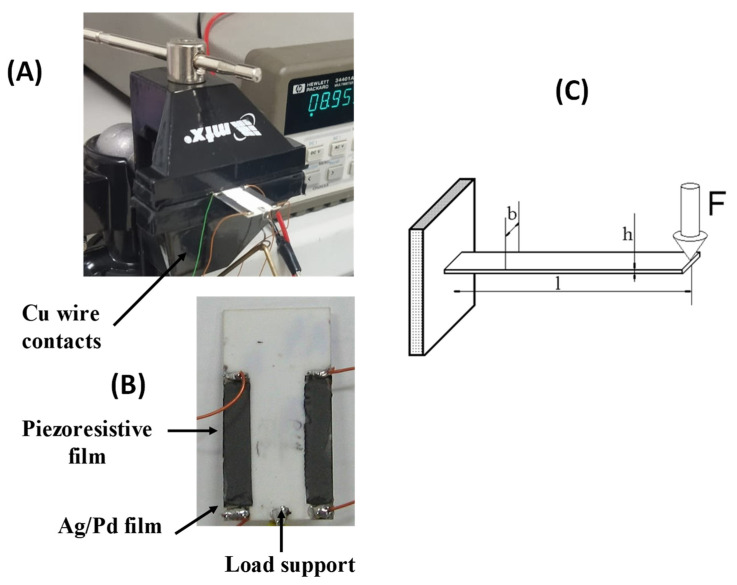
Experimental set up for measurements of electrical response under strain deformation: (**A**) Fixation of a sensor for strain tests, (**B**) Sensor view, (**C**) Schematic of the cantilever clamped beam geometry.

**Figure 3 sensors-22-03256-f003:**
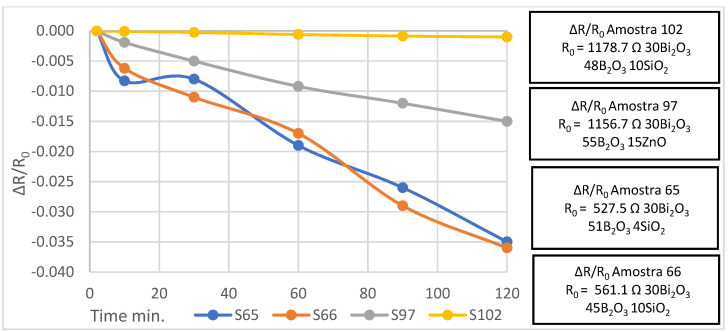
Results of resistance evolution after strain load removal.

**Figure 4 sensors-22-03256-f004:**
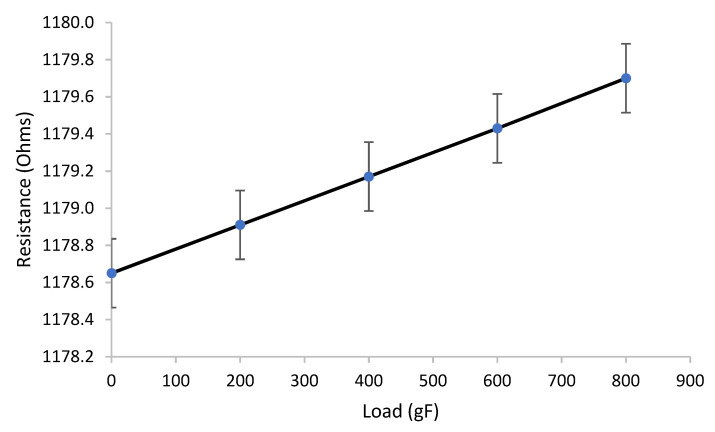
Prototype device response under strain test, sample 102.

**Figure 5 sensors-22-03256-f005:**
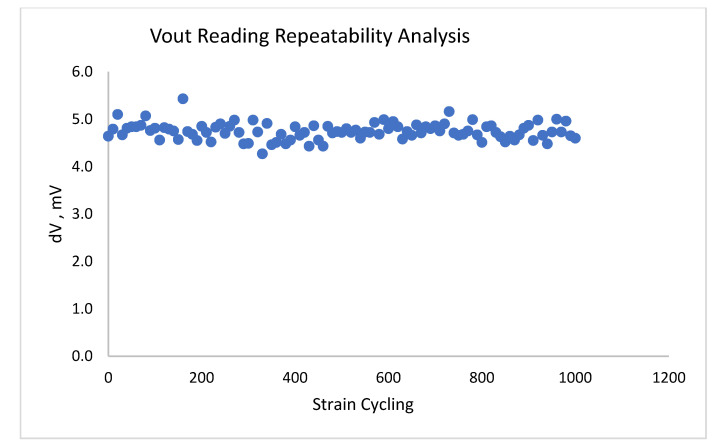
Results of repeatability tests, 1000 cycles.

**Figure 6 sensors-22-03256-f006:**
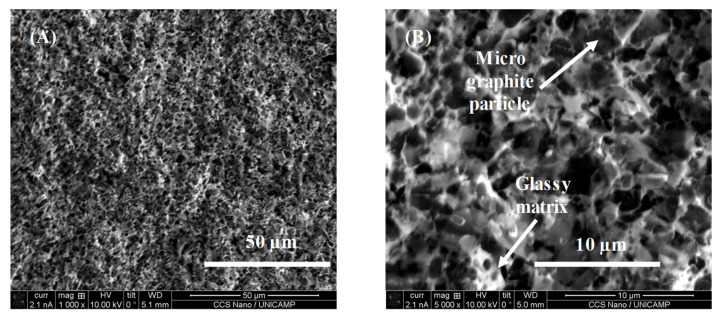
Scanning electronic microscopy image of the piezoresistive film surface, sample 102. Scale bars 50 µm (**A**) and 10 µm (**B**). Arrows show the positions of micrographite particles (top) and glassy matrix (bottom).

**Figure 7 sensors-22-03256-f007:**
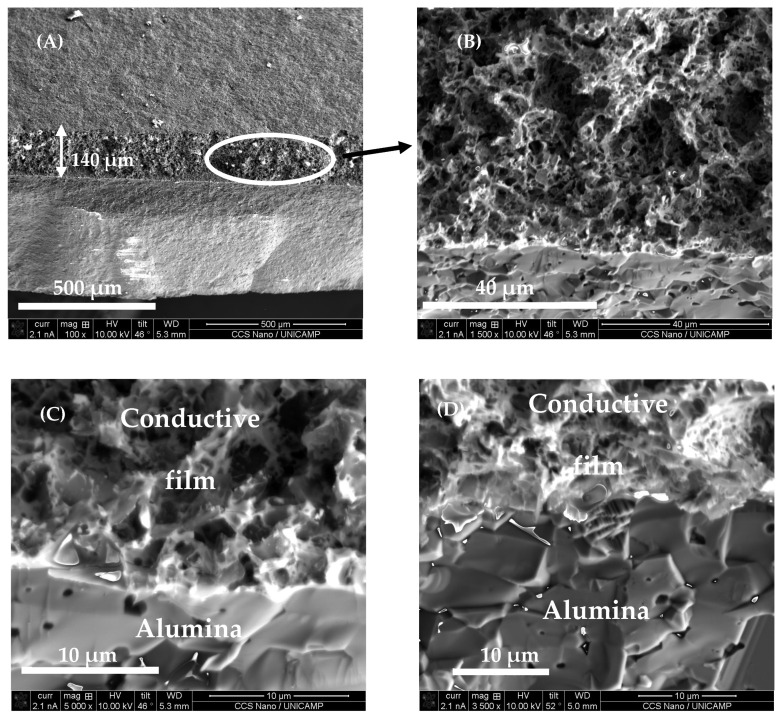
Scanning electronic microscopy image of the fractured piezoresistive film cross-section (**A**,**B**) and the interface between conductive film and alumina (**C**,**D**). Scale bars 500 µm (**A**), 40 µm (**B**), 10 µm (**C**,**D**).

**Figure 8 sensors-22-03256-f008:**
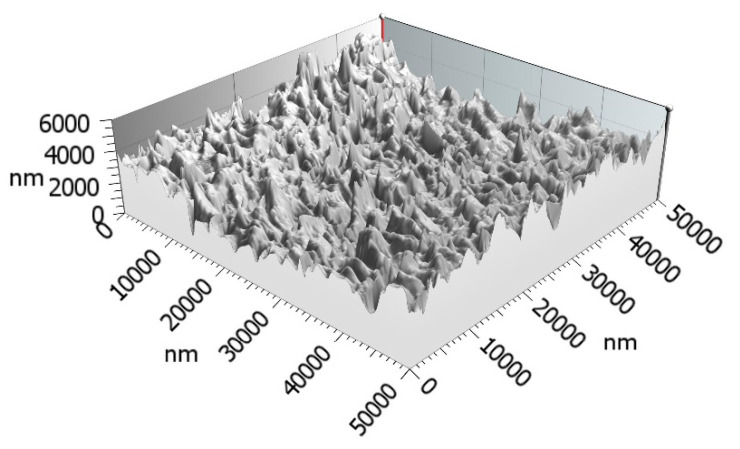
3D image of the piezoresistive film surface, with vertical and lateral scales of 6000 nm and 50,000 nm (6 and 50 µm), respectively.

**Figure 9 sensors-22-03256-f009:**
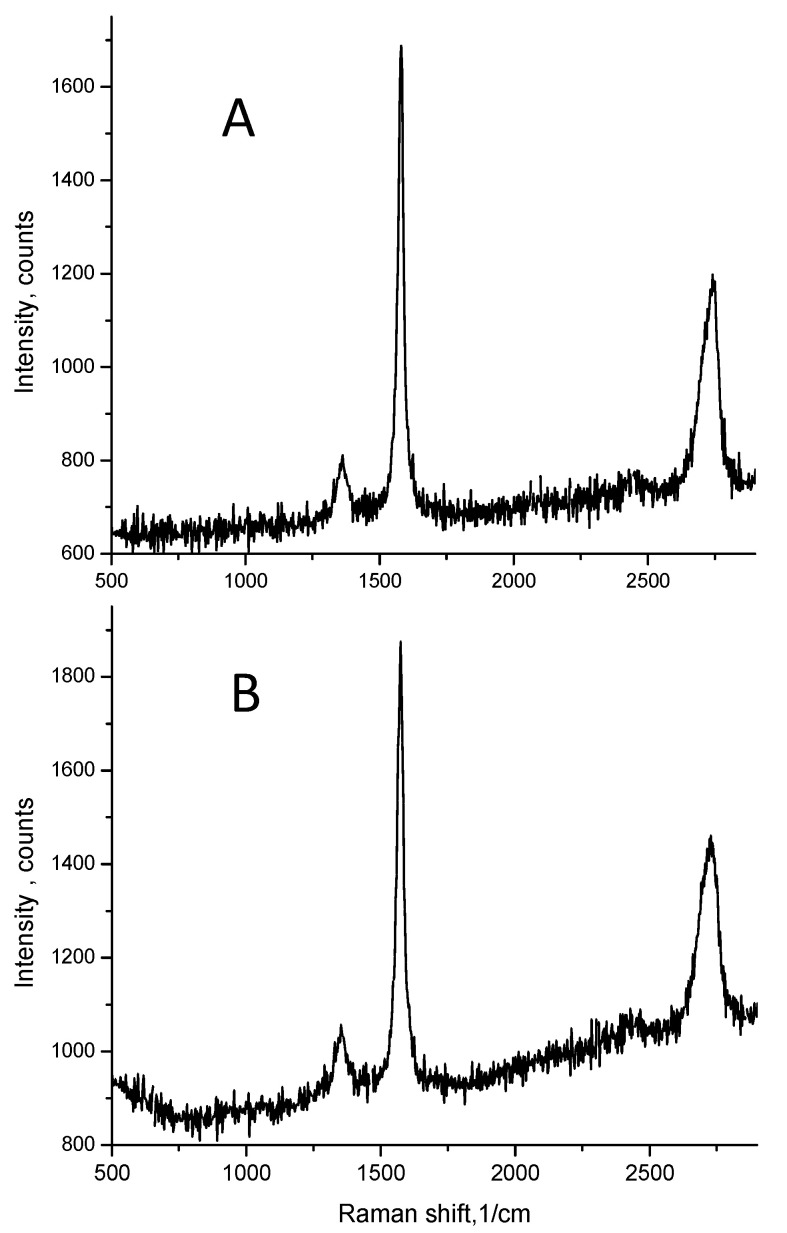
Raman spectra of the micrographite (**A**) and fabricated piezoresistive film (**B**), from 500 to 2800 cm^−1^.

**Table 1 sensors-22-03256-t001:** Glass compositions.

Sample	Bismuth Oxide	Boron Oxide	Zinc Oxide	Silicon Oxide	Aluminum Oxide
97	30Bi_2_O_3_	55B_2_O_3_	15ZnO		
65	30Bi_2_O_3_	51B_2_O_3_	15ZnO	4SiO_2_	
66	30Bi_2_O_3_	45B_2_O_3_	15ZnO	10SiO_2_	
102	30Bi_2_O_3_	48B_2_O_3_	10ZnO	10SiO_2_	2Al_2_O_3_

## Data Availability

The datasets generated during and/or analyzed during the current study are available from the corresponding author on reasonable request.

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
