# Peer review of "Piezoresistive Sensor Based on Micrographite-Glass Thick Films"

_sensors, 2022, doi:10.3390/s22093256_

Round 1

Reviewer 1 Report

the authors report Piezoresistive Sensor Based on Micrographite - glass Thick Films. This paper can be considered for publication after some modifications:
1. the error bars should be added in all figures
2. The introduction should be rewritten with more recent references 
3. In Fig 7 and Fig 8, the scale of the images is invisible, the resolution of this figure should be improved

Author Response

The authors thank the reviewer for valuable comments,, changes were done as follows:

  1. the error bars should be added in all figures - Done
  2. The introduction should be rewritten with more recent references - Done, new Refs. 6-11 added.
  3. In Fig 7 and Fig 8, the scale of the images is invisible, the resolution of this figure should be improved - The scales are shown, the resolution of figures is maximum possible.

Reviewer 2 Report

Journal                 Sensors (ISSN 1424-8220)

Manuscript ID    sensors-1638887

Type                      Article

Title                       Piezoresistive Sensor Based on Micrographite-glass Thick Films

Authors                Osvaldo Correa , Pompeu Pereira de Abreu Filho , Stanislav Moshkalev * , Jacobus Willibrordus Swart

Section                 Sensor Materials

Special Issue      Graphene-Based Strain and Pressure Sensors

Summary: Author has developed and demonstrated an Piezoresistive sensor. Sensors has been developed in Pb-free glass containing several oxides.

Comments:

  • The abstract should include existing challenges and issues resolved in the present research.
  • The author should share the range in which sensors have been tested.
  • Line 49: authors directly introduced the present case. The author should elaborately present the motivation and gaps behind the development of the present sensor.
  • Line 57: Very few articles from previous reports are discussed to reach to a conclusion in the introduction section. The author should present more literature.
  • The last paragraph of the introduction section should present details of the present research. The existing last paragraph should be moved up in the introduction section.
  • The reproducibility of experiments at other labs is of prime importance. The author should share the procedure; percentages of materials uses and detailed synthesis procedure. Few steps in Materials and Methods need to be supplemented with more details.
  • As far as possible author should use SI units only.
  • If journal permits, the Author should add a conclusions section too.

Author Response

The authors thank the Reviewer for valuable comments and suggestions.

Corrections to the text were added accordingly:

 The abstract should include existing challenges and issues resolved in the present research.

  • The main challenge was to replace extremely expensive ruthenium oxide as a conductive phase by low cost micrographite , this was added to the abstract

The author should share the range in which sensors have been tested.

  • The range of deflections at the cantilever edge was added (line 140).

Line 49: authors directly introduced the present case. The author should elaborately present the motivation and gaps behind the development of the present sensor.

  • The motivation was more elaborated,

Line 57: Very few articles from previous reports are discussed to reach to a conclusion in the introduction section. The author should present more literature.

  • Recent publications [6-11] were added.

The last paragraph of the introduction section should present details of the present research. The existing last paragraph should be moved up in the introduction section.

The reproducibility of experiments at other labs is of prime importance. The author should share the procedure; percentages of materials uses and detailed synthesis procedure. Few steps in Materials and Methods need to be supplemented with more details.

  • More information added.

As far as possible author should use SI units only.

  • SI units are used when appropriate

If journal permits, the Author should add a conclusions section too.

  • Conclusion added.

Reviewer 3 Report

The paper needs some improvement.

In general: All figures should be improved! Fig 2 a and b should be larger.

                 Text should be smaler (Fig 5, Fig 4)

Page 6: Fig 4.: What is the temeprature dependence?

             On Fig. 5 dots shoul be smaller.

Page 9: Where is conclusion?

Page 1: Introduction: Line 33: Piezoresistive sensors are important also in the oscillator methods which have high sensitivity and accuracy such as they are described in ref.:

  • Matko, V., Milanovič, M. Detection principles of temperature compensated oscillators with reactance influence on piezoelectric resonator. Sensors. 2020, vol. 20, iss. 3, p. 1-18. ISSN 1424-8220. https://www.mdpi.com/1424-8220/20/3/802
  • -Yang, S.; Tan, M.; Yu, T.; Li, X.; Wang, X.; Zhang, J. Hybrid Reduced Graphene Oxide with Special Magnetoresistance for Wireless Magnetic Field Sensor. Nano-Micro Letters 2020, 12, 1-14, doi:10.1007/s40820-020-0403-9.

Authors should include mentioned references in the text.

Author Response

The authros thank the reviewer for comments , changes were done in the text as follows:

In general: All figures should be improved! Fig 2 a and b should be larger. - Done

Text should be smaler (Fig 5, Fig 4) - Done.

Page 6: Fig 4.: What is the temperature dependence? The temperature dependence was not studied at this stage, will be the subject of future work.

On Fig. 5 dots shoul be smaller. - Done

Page 9: Where is conclusion? - Conclusion section is added.

Page 1: Introduction: Line 33: Piezoresistive sensors are important also in the oscillator methods which have high sensitivity and accuracy such as they are described in ref.:

  • Matko, V., Milanovič, M. Detection principles of temperature compensated oscillators with reactance influence on piezoelectric resonator. Sensors. 2020, vol. 20, iss. 3, p. 1-18. ISSN 1424-8220. https://www.mdpi.com/1424-8220/20/3/802
  • -Yang, S.; Tan, M.; Yu, T.; Li, X.; Wang, X.; Zhang, J. Hybrid Reduced Graphene Oxide with Special Magnetoresistance for Wireless Magnetic Field Sensor. Nano-Micro Letters 2020, 12, 1-14, doi:10.1007/s40820-020-0403-9.

Authors should include mentioned references in the text. - References. are added

Round 2

Reviewer 1 Report

none

Author Response

The reveiwer did not ask for any revision

Reviewer 2 Report

ournal Sensors (ISSN 1424-8220) Manuscript ID sensors-1638887 Type Article Title Piezoresistive Sensor Based on Micrographite-glass Thick Films Authors Osvaldo Correa , Pompeu Pereira de Abreu Filho , Stanislav Moshkalev * , Jacobus Willibrordus Swart Section Sensor Materials   Accepted

Author Response

Revision is done